# ChatGPT Beyond English: Towards a Comprehensive Evaluation of Large Language Models in Multilingual Learning

**Viet Dac Lai**[1*], **Nghia Trung Ngo**[1*], **Amir Pouran Ben Veyseh**[1*],
**Hieu Man**[1], **Franck Dernoncourt**[2], **Trung Bui**[2], **Thien Huu Nguyen**[1]

[1]Dept. of Computer Science, University of Oregon, OR, USA
[2]Adobe Research, USA

{vietl@cs,nghian,apouranb@cs,hieum,thien@cs}@uoregon.edu
{franck.dernoncourt,bui}@adobe.com

## Abstract

Over the last few years, large language models (LLMs) have emerged as the most important breakthroughs in natural language processing (NLP) that fundamentally transform research and developments in the field. ChatGPT represents one of the most exciting LLM systems developed recently to showcase impressive skills for language generation and highly attract public attention. Among various exciting applications discovered for ChatGPT in English, the model can process and generate texts for multiple languages due to its multilingual training data. Given the broad adoption of ChatGPT for English in different problems and areas, a natural question is whether ChatGPT can also be applied effectively for other languages or it is necessary to develop more language-specific technologies. The answer to this question requires a thorough evaluation of ChatGPT over multiple tasks with diverse languages and large datasets (i.e., beyond reported anecdotes), which is still missing or limited in current research. Our work aims to fill this gap for the evaluation of ChatGPT and similar LLMs to provide more comprehensive information for multilingual NLP applications. In particular, we evaluate ChatGPT on 7 different tasks, covering 37 diverse languages with high, medium, low, and extremely low resources. Compared to the performance of previous models, our extensive experiments demonstrate the worse performance of ChatGPT for different NLP tasks and languages, calling for further research to develop better models and understanding for multilingual learning.

## 1 Introduction

Since the introduction of word embeddings (Bengio et al., 2000) and deep learning architectures (Collobert et al., 2011), Natural Language Processing (NLP) has witnessed significant breakthroughs that fundamentally transform research and applications in various areas. Starting with the creation of word2vec (Mikolov et al., 2013), the major milestones in NLP involve the presentation of the seq2seq or encoder-decoder framework (Cho et al., 2014; Sutskever et al., 2014), the proposal of the attention mechanism (Bahdanau et al., 2015), the development of Transformer architecture (Vaswani et al., 2017), the notion of uncontextualized word embeddings from language models in ELMo (Peters et al., 2018), and the pre-trained transformer-based language models, e.g., BERT (Devlin et al., 2019), GPT (Radford et al., 2018, 2019), T5 (Raffel et al., 2020), and BART (Lewis et al., 2020).

The recent advances in NLP feature large language models (LLMs) that have parameter sizes over a hundred billion and are pre-trained on massive data, e.g., GPT-3 (Rae et al., 2021), Megatron (Shoeybi et al., 2019), GPT-Jurassic (Lieber et al., 2021), OPT-175B (Zhang et al., 2022b), and multilingual BLOOM (Scao et al., 2022). Although still relying on the Transformer architecture, the unprecedented scales of model size and training data have allowed new emergent abilities to change the landscape and practices in NLP (Wei et al., 2022). An important emergent skill involves prompt-based learning that facilities the probing of information from LLMs with prompts by sampling the learned language distributions (Brown et al., 2020). In this way, the models demonstrate strong generalization in few-shot and zero-shot learning while avoiding parameter updates for the underlying architectures.

To this end, ChatGPT[1] is one of the latest developments in NLP. In the first two months of its launch, ChatGPT has attracted 100 million users (Milmo, 2023). As the next iteration of InstructGPT (Ouyang et al., 2022), ChatGPT is optimized on top of a GPT-3.5 series model using reinforcement learning from human feedback (RLHF) (Christiano et al., 2017). In contrast to pre-

---

* The first three authors contributed equally to this work.

[1]https://openai.com/blog/chatgpt/

vious LLMs, ChatGPT and InstructGPT leverage human demonstrations of desired outputs for input prompts to train supervised models, while human rankings of generated outputs are obtained to train a reward model to further optimize the LLMs with reinforcement learning. Compared to InstructGPT, ChatGPT is trained with conversational data to allow follow-up questions. In this way, ChatGPT is able to interact with humans in multi-turn conversations to generate more aligned outputs with human interests, thus being more natural and accessible to users. In addition, due to the deployment of public APIs to facilitate general users, there have been multiple reports on the successes of ChatGPT in solving challenging tasks in various areas, e.g., passing the United States Medical Licensing Examination (Kung et al., 2022) and real exams in a law school (Choi et al., 2023), performing competitively with commercial translation services for some high-resource languages (Jiao et al., 2023), and even producing code from natural language instructions. Nonetheless, the communities also express concerns about long-term implications of ChatGPT and LLMs for society, citing issues on plagiarism, privacy, misinformation, and security (Bang et al., 2023).

Similar to other LLMs, ChatGPT is trained on a mix of training data from multiple languages. Although English is the majority, the combination of multilingual data contributes to ChatGPT's abilities to accept inputs and generate responses in different languages, making it accessible and widely adopted by people around the world. However, given the recency of the technology, ChatGPT has been mainly evaluated over English data. The community is lacking a comprehensive, public, and independent evaluation of ChatGPT over various non-English languages for diverse NLP tasks to provide proper perspectives for future research and applications. Given ChatGPT's transformative potentials, associated long-term risks, huge cost for training, and limited transparency, a fundamental question is whether multilingual LLMs such as ChatGPT can also be reliably adopted for different languages or it is necessary to develop language-specific LLMs/other technologies to solve NLP problems for non-English languages.

To address the multilingual concerns for Chat-GPT, a few recent studies have investigated ChatGPT's performance and responses for non-English languages. However, the considered tasks/languages/settings and scale of evaluation data in existing multilingual evaluations are still limited, which is unable to show a comprehensive picture of the potentials/performance of the technology on a diversity of other languages. For instance, (Bang et al., 2023) evaluates the multilingual performance of ChatGPT on three tasks of language identification, sentiment analysis, and machine translation; however, only a few languages are selected for each task and the number of evaluation samples for each language does not exceed 50. Beyond English, the analysis of ChatGPT's responses for input questions in (Guo et al., 2023) is only done for Chinese, while the results of the medical licensing examinations for ChatGPT are only shown for Japanese in (Kasai et al., 2023). In addition, (Fang et al., 2023) and (Wang et al., 2023a) explores ChatGPT in three languages English, Chinese, and German; however, the studies only focus on grammatical error correction or cross-lingual summarization.

To this end, our paper aims to perform a more thorough evaluation of ChatGPT for its performance on multiple languages over different NLP tasks. Our experiments consider 37 diverse languages, characterizing *high-, medium-, low-, and extremely low-resource languages*, to better highlight ChatGPT's potentials and limitations. To our knowledge, this is one of the largest sets of languages evaluated for ChatGPT in a public study to date. In addition to Natural Language Inference (NLI), Question Answering, and Common Sense Reasoning, our current work will examine the tasks of Part-of-Speech (POS) Tagging, Named Entity Recognition (NER), Relation Extraction, and Summarization, which are not covered in previous multilingual evaluations for ChatGPT. To improve the reproducibility of the evaluations and better reflect the approach of general users, our current work will focus on the zero-shot learning setting for ChatGPT where no human-provided examples are presented to the model. Importantly, due to the scale of available languages/tasks/datasets/models and the growing nature of multilingual learning research in NLP, we will use this work as an ongoing and public effort to evaluate ChatGPT and other LLMs for multiple languages, emphasizing on understudied languages to measure robustness and democratize impacts of the technologies. Despite some potential updates with future experiments, our current experiments suggest the following tendencies:

- ChatGPT's zero-shot learning performance is generally worse than the state-of-the-art performance of the supervised learning models for a majority of the considered tasks across different languages, including high-, medium-, low-, and extremely-low resource languages. The performance gaps are usually very large, demonstrating the unfit of ChatGPT as a general solver for different NLP problems. It thus highlights the importance of task-specific models for the development of NLP applications.

- ChatGPT's performance is generally better for English than for other languages, especially for higher-level tasks that require more complex reasoning abilities (e.g., named entity recognition, question answering, common sense reasoning, and summarization). The performance differences can be substantial for some tasks and lower-resource languages, which justifies the biases of ChatGPT for English and suggests the necessity to develop language-specific models/LLMs for different languages and groups.

- ChatGPT can perform better with English prompts even though the task and input texts are intended for other languages, further confirming the biases toward English of ChatGPT.

## 2 Related Work

Since the release of ChatGPT in November 2022 with impressive language abilities, there has been a growing interest in evaluating ChatGPT for different aspects of natural language understanding. The first line of work concerns the performance comparison of ChatGPT and state-of-the-art systems for important tasks in NLP such as text summarization (Wang et al., 2023a; Yang et al., 2023), machine translation (Hendy et al., 2023; Jiao et al., 2023; Kocmi and Federmann, 2023), question answering (Tan et al., 2023; Omar et al., 2023; Lai et al., 2023), information extraction (Wei et al., 2023; Gao et al., 2023), text classification (Kuzman et al., 2023; Amin et al., 2023), grammatical error detection (Fang et al., 2023), and stance detection (Zhang et al., 2022a). Along this line, several recent studies have attempted to examine the performance of ChatGPT more comprehensively on multiple datasets (Bang et al., 2023; Qin et al., 2023; Koco'n et al., 2023; Zhong et al., 2023). The second direction for ChatGPT evaluation focuses on

the robustness/reliability of the model against possible variants of input texts. For example, (Wang et al., 2023b) explores the robustness of ChatGPT under the adversarial and out-of-domain learning settings while (Jang and Lukasiewicz, 2023) examines the logical prediction consistency of ChatGPT for inputs with semantic equivalence, logical negation, or symmetricity. Finally, the third dimension for ChatGPT evaluation discusses the potential impacts and risks of the technology for the broader society, e.g., in education (Susnjak, 2022; Khalil and Er, 2023), law (Choi et al., 2023), medical (Kung et al., 2022), ethnics (Shen et al., 2023), human-computer collaboration (Lanzi and Loiacono, 2023), and cognition (Mahowald et al., 2023). However, to our knowledge, none of existing work has conducted large-scale evaluations of ChatGPT for multiple and diverse languages/tasks as we do.

## 3 Methodology

The goal of our research is to evaluate the performance of ChatGPT and LLMs for NLP tasks in different languages. Given the large numbers of NLP datasets/tasks/languages and the growing developments of LLMs, our work will be an ongoing effort to include additional experiments to be more comprehensive along the way. In the current version of the paper, we will evaluate ChatGPT on seven diverse NLP tasks, i.e., Part-of-Speech (POS) Tagging, Named Entity Recognition (NER), Relation Classification, Natural Language Inference (NLI), Question Answering (QA), Common Sense Reasoning (CSR), and Summarization. Over different tasks, our experiments will cover 34 diverse languages, characterizing high-, medium-, low-, and extremely low-resource languages to provide broader perspectives. Following (Bang et al., 2023), we employ the ratio of the data for each language in the CommonCrawl corpus[2], i.e., the main data to pre-train GPT-3, to classify the resource levels. In particular, a language will be considered as high-, medium-, low-, and extremely low-resource if its data ratio is greater than $1\%$ ($> 1\%$), between $0.1\%$ and $1\%$ ($> 0.1\%$), between $0.01\%$ and $0.1\%$ ($> 0.01\%$), and smaller than $0.01\%$ ($< 0.01\%$) respectively. Table 1 presents information and categories for the languages considered in our work.

As the scale of ChatGPT precludes the ability to fine-tune the model on downstream task data for most general users, we focus on the zero-shot

---

[2] http://commoncrawl.org

| Language | Code | Pop. (M) | CC Size (%) | Cat. |
|---|---|---|---|---|
| English | en | 1,452 | 45.8786 | H |
| Russian | ru | 258 | 5.9692 | H |
| German | de | 134 | 5.8811 | H |
| Chinese | zh | 1,118 | 4.8747 | H |
| Japanese | jp | 125 | 4.7884 | H |
| French | fr | 274 | 4.7254 | H |
| Spanish | es | 548 | 4.4690 | H |
| Italian | it | 68 | 2.5712 | H |
| Dutch | nl | 30 | 2.0585 | H |
| Polish | pl | 45 | 1.6636 | H |
| Portuguese | pt | 257 | 1.1505 | H |
| Vietnamese | vi | 85 | 1.0299 | H |
| Turkish | tr | 88 | 0.8439 | M |
| Indonesian | id | 199 | 0.7991 | M |
| Swedish | sv | 13 | 0.6969 | M |
| Arabic | ar | 274 | 0.6658 | M |
| Persian | fa | 130 | 0.6582 | M |
| Korean | ko | 81 | 0.6498 | M |
| Greek | el | 13 | 0.5870 | M |
| Thai | th | 60 | 0.4143 | M |
| Ukrainian | uk | 33 | 0.3304 | M |
| Bulgarian | bg | 8 | 0.2900 | M |
| Hindi | hi | 602 | 0.1588 | M |
| Bengali | bn | 272 | 0.0930 | L |
| Tamil | ta | 86 | 0.0446 | L |
| Urdu | ur | 231 | 0.0274 | L |
| Malayalam | ml | 36 | 0.0222 | L |
| Marathi | mr | 99 | 0.0213 | L |
| Telugu | te | 95 | 0.0183 | L |
| Gujarati | gu | 62 | 0.0126 | L |
| Burmese | my | 33 | 0.0126 | L |
| Kannada | kn | 64 | 0.0122 | L |
| Swahili | sw | 71 | 0.0077 | X |
| Punjabi | pa | 113 | 0.0061 | X |
| Kyrgyz | ky | 5 | 0.0049 | X |
| Odia | or | 39 | 0.0044 | X |
| Assamesese | as | 15 | 0.0025 | X |

Table 1: List of languages, language codes, numbers of first and second speakers, data ratios in the CommonCrawl corpus, and language categories. The languages are grouped into categories based on their data ratios in the CommomCrawl corpus: High Resource (H, $> 1\%$), Medium Resource (M, $> 0.1\%$), and Low Resource (L, $> 0.01\%$), and Extremely-Low Resource (X, $< 0.01\%$).

learning setting for ChatGPT. We also report the state-of-the-art performance of the supervised models for a task in each language as a reference for research progress. In zero-shot learning, an NLP task $T$ is specified by a natural-language task description $D$. Given a new data sample with input text $X$ for the task $T$, the concatenation of $D$ and $X$ will then be sent into the ChatGPT model $G$ as the input prompt to generate a natural-language response $R = G([D; X])$. Afterward, the response $R$ will be parsed using pre-defined task-specific rules $P$ to obtain an output $Y = P(R(G([D; X]))$ in the required format for $T$ (e.g., a pre-defined label for classification problems). Finally, the outputs $Y$ for examples in an evaluation dataset will be scored to return ChatGPT's performance for task $T$.

Different from some previous work that exploits two-stage prompting to adopt a zero-shot chain of thoughts (Kojima et al., 2022; Qin et al., 2023), we directly utilize single-stage prompting that only adds the task description $D$ into each input $X$ to simulate the common approach of general users for ChatGPT. Other prompting strategies can be explored in future work. As such, in the current version, we aim to design simple task descriptions $D$ while ensuring necessary information to indicate the task and facilitate the parsing of responses to produce accurate outputs $Y$. In addition, for tasks in a non-English target language, we will evaluate task descriptions in both English and target-specific languages to shed light on the best approach to prompt ChatGPT in multilingual settings. To facilitate the experiments, all non-English task descriptions are obtained using the automatic translation tool Google Translate[3] to translate the designed English descriptions for each task. Finally, all of the responses from ChatGPT in this work are obtained between March 1 and April 5. This is right after ChatGPT is made available in OpenAI APIs to enable large-scale requests from the public for comprehensive evaluations. To improve reproducibility, we clear the conversations in ChatGPT for each query to remove any previous context. In the following, due to the space constraint, we will only describe the tasks, datasets, and ChatGPT's performance. The designed prompts for each task will be provided in the Appendix.

## 4 Part-of-Speech Tagging

Part-of-Speech (POS) Tagging is a coarse-grained word classification task whose goal is to label the syntactic information of the words in a sentence.

---

[3] https://translate.google.com

We evaluate ChatGPT for its multilingual POS tagging abilities over the XGLUE-POS dataset (Liang et al., 2020), which covers 18 languages and includes labels derived from the Universal Dependencies (UD) Treebanks (v2.5) (Zeman et al., 2020). In the experiments, we utilize the XGLUE-POS dataset from Huggingface Datasets[4] that only includes 17 languages (e.g., excluding Portuguese). As such, we use the test sets of XGLUE-POS with more than 15K samples for the selected languages in the evaluation. Appendix A provides details for our POS Tagging prompt for ChatGPT.

| Language | Code | Cat. | XLM-R | ChatGPT | |
| --- | --- | --- | --- | --- | --- |
| | | | | (en) | (spc) |
| English | en | H | 96.2 | 88.5 | 89.6 |
| Russian | ru | H | 86.9 | 91.6 | 59.1 |
| German | de | H | 92.2 | 90.2 | 89.9 |
| Chinese | zh | H | 60.4 | 76.5 | 75.3 |
| French | fr | H | 89.9 | 93.2 | 93.5 |
| Spanish | es | H | 89.0 | 92.2 | 91.9 |
| Italian | it | H | 92.6 | 92.6 | 93.4 |
| Dutch | nl | H | 88.5 | 88.1 | 88.3 |
| Polish | pl | H | 85.4 | 90.4 | 64.5 |
| Vietnamese | vi | H | 55.2 | 64.8 | 65.9 |
| Turkish | tr | M | 72.7 | 78.6 | 69.6 |
| Arabic | ar | M | 67.3 | 81.0 | 80.9 |
| Greek | el | M | 88.2 | 87.1 | 79.8 |
| Thai | th | M | 57.9 | 68.5 | 69.1 |
| Bulgarian | bg | M | 88.8 | 91.2 | 92.3 |
| Hindi | hi | M | 74.5 | 83.1 | 72.8 |
| Urdu | ur | L | 62.1 | 78.4 | 80.7 |
| Average | | | 79.3 | 84.5 | 79.8 |

Table 2: Accuracy of ChatGPT (zero-shot learning) and XLM-R (supervised learning) on the test sets of XGLUE-POS. ChatGPT is evaluated with both English (en) and language-specific (spc) task descriptions.

**Results:** Table 2 presents the performance of ChatGPT (zero-shot learning with both English and language-specific task descriptions) and the fully supervised XLM-R model (based on XLM-RoBERTa base) (Liang et al., 2020). Here, performance is measured via the accuracy of the predicted POS tags. As can be seen, ChatGPT outperforms XLM-R over 13 out 17 languages for multilingual POS tagging. Different from XLM-R where English has the best POS tagging performance, ChatGPT seems to have better accuracy than English with some other languages (e.g., French, Spanish). Finally, we observe that English prompts tend to perform better or at lest competi-

---

[4]https://huggingface.co/datasets/xglue

tively with language-specific prompts for ChatGPT across different languages for POS tagging.

## 5   Named Entity Recognition

Named Entity Recognition (NER) is an important task in NLP (Sang and Meulder, 2002), aiming to identify spans and semantic types of names (e.g., person, organization) in text. NER is usually formulated as a sequence tagging problem where a label is assigned to each word in a sentence to indicate names. The BIO annotation schema is often leveraged to form the labels to capture both span and type information (Ratinov and Roth, 2009). For multilingual NER evaluation of ChatGPT, we employ the datasets from the recent shared task MultiCoNER (Malmasi et al., 2022) that seeks to build NER systems for 11 languages following the WNUT 2017 taxonomy for entity types (Derczynski et al., 2017). There are 6 entity types in Multi-CoNER, i.e., PER (person), LOC (location), CORP (corporation), CW (creative work), GRP (group of people), and PROD (product). We utilize the test sets of the language in MultiCoNER for evaluation. Our prompt for the NER task is described in Appendix B.

| Language | Code | Cat. | DAMO | ChatGPT | |
| --- | --- | --- | --- | --- | --- |
| | | | | (en) | (spc) |
| English | en | H | 91.2 | 37.2 | 37.2 |
| Russian | ru | H | 91.5 | 27.4 | 22.0 |
| German | de | H | 90.7 | 37.1 | 32.8 |
| Chinese | zh | H | 81.7 | 18.8 | 19.8 |
| Spanish | es | H | 89.9 | 34.7 | 33.2 |
| Dutch | nl | H | 90.5 | 35.7 | 37.5 |
| Turkish | tr | M | 88.7 | 31.9 | 29.1 |
| Persian | fa | M | 89.7 | 25.9 | 21.9 |
| Korean | ko | M | 88.6 | 30.0 | 32.2 |
| Hindi | hi | M | 86.2 | 27.3 | 26.1 |
| Bengali | bn | L | 84.2 | 23.3 | 16.4 |
| Average | | | 88.4 | 29.9 | 28.0 |

Table 3: Performance (F1 scores) of ChatGPT (zero-shot learning) and DAMO (supervised learning) on the test sets of MultiCoNER. ChatGPT is evaluated with both English (en) and language-specific (spc) task descriptions.

**Results**: Table 3 evaluates the performance of Chat-GPT (zero-shot learning with both English and language-specific task descriptions) and DAMO (Wang et al., 2022a), the model with current best-reported performance on MultiCoNER. The latter retrieves relevant context from Wikipeida for each

input sentence that are then fed into the XLMR-RoBERTa model (large version) for NER. DAMO also employ a conditional random fields (CRF) layer for the modeling. Our results for NER are evaluated using macro-averaged F1 scores (Malmasi et al., 2022). The most important observation from the table is that ChatGPT significantly underperforms DAMO on MultiCoNER across all 11 languages. In fact, the performance of ChatGPT is less than 40% for all languages, which suggests less suitability of ChatGPT to solve NER in this domain. Finally, we provide more analysis for the performance of ChatGPT for NER in Appendix C.

# 6 Relation Extraction

Relation Extraction (RE) is a crucial task in information extraction (IE), aiming to identify and classify semantic relations between two entity mentions in an input text. To facilitate multilingual experiments for RE, we conduct our evaluation over the SMiLER dataset (Seganti et al., 2021). SMiLER provides relation annotation for texts in 14 languages with 36 relation types (including "*norelation*"). The test sets of the languages (with more than 12K samples) are employed for evaluation. We present our ChatGPT prompt for RE in Appendix D.

| Language | Code | Cat. | mT5-IL | ChatGPT (en) | ChatGPT (spc) |
|---|---|---|---|---|---|
| English | en | H | 96.0 | 61.9 | 61.8 |
| Russian | ru | H | 83.3 | 78.8 | 77.5 |
| German | de | H | 94.0 | 71.1 | 71.8 |
| French | fr | H | 97.2 | 72.4 | 73.9 |
| Spanish | es | H | 70.5 | 67.5 | 65.8 |
| Italian | it | H | 97.0 | 74.4 | 74.6 |
| Dutch | nl | H | 93.5 | 66.8 | 66.6 |
| Polish | pl | H | 93.0 | 63.4 | 65.8 |
| Portuguese | pt | H | 85.2 | 64.8 | 66.3 |
| Arabic | ar | M | 94.1 | 84.9 | 90.1 |
| Persian | fa | M | 73.1 | 58.9 | 63.8 |
| Korean | ko | M | 83.2 | 65.3 | 70.1 |
| Swedish | sv | M | 58.7 | 64.2 | 65.4 |
| Ukrainian | uk | M | 71.8 | 76.5 | 68.8 |
| Average | | | 85.0 | 69.4 | 70.2 |

Table 4: Performance (F1 scores) of ChatGPT (zero-shot learning) and mT5-IL (supervised learning) on the test sets of SMiLER. ChatGPT is evaluated with both English (en) and language-specific (spc) task descriptions.

**Results:** Table 4 shows the performance of Chat-GPT (zero-shot learning with both English and language-specific task descriptions) and mT5-IL (Chen et al., 2022), a state-of-the-art supervised in-language prompting model for SMiLER. mT5-IL is based on the base version of mT5. Micro F1 scores are used as the performance metric for RE. From Table 4, the results suggest that mT5-IL significantly outperforms ChatGPT over different languages no matter if we ask ChatGPT with English or language-specific prompts (except for Swedish and Ukranian). The performance gap is up to 15% over F1 score on average for the languages. Language-specific prompts seem to yield better or comparable performance as English prompts for ChatGPT with RE. Ukrainian is an exception when English prompts return better F1 score for Chat-GPT. Interestingly, ChatGPT performs the worst in English for RC with SMiLER, potentially due to the much larger size of English test data with greater diversity and challenges (5,461 samples for English vs. 1,243 samples for the second large test set for French).

# 7 Natural Language Inference

Natural Language Inference (NLI) aims to predict the entailment/contradiction relations between two input sentences, i.e., a premise and a hypothesis. To evaluate ChatGPT for multilingual NLI, we utilize the XNLI dataset (Conneau et al., 2018) that provides annotated data for English and 14 other languages with three categories, i.e., *Entailment*, *Contradiction*, and *Neutral*. As such, the data in non-English languages is obtained by translating English data for XNLI. XNLI provides development and test data to facilitate development and evaluation. However, as the labels for the test data are not publicly available, we utilize the development data of XNLI in this experiment. The Chat-GPT prompt for NLI is described in Appendix E.

**Results**: Table 5 reports the performance (accuracy) of ChatGPT and the multilingual model mT5-XXL (Xue et al., 2021). Here, for each non-English target language, we present ChatGPT's performance on two zero-shot learning settings depending on whether the task descriptions are in English or target language. For mT5-XXL, the model is fine-tuned on English training data and translations in the target language to achieve the best reported performance on XNLI. It is clear from the table that ChatGPT performs significantly poorer than mT5-XXL across different languages by large margins. The performance gaps between ChatGPT

| Language | Code | Cat. | mT5-XXL | ChatGPT (en) | ChatGPT (spc) |
|---|---|---|---|---|---|
| English | en | H | 92.4 | 70.2 | 70.2 |
| Russian | ru | H | 86.4 | 60.8 | 45.4 |
| German | de | H | 89.2 | 64.5 | 51.1 |
| Chinese | zh | H | 86.2 | 58.2 | 35.5 |
| French | fr | H | 88.7 | 64.8 | 42.2 |
| Spanish | es | H | 89.4 | 65.8 | 47.4 |
| Vietnamese | vi | H | 86.6 | 55.4 | 44.8 |
| Turkish | tr | M | 86.4 | 57.1 | 37.1 |
| Arabic | ar | M | 87.1 | 55.3 | 22.3 |
| Greek | el | M | 88.7 | 55.9 | 54.5 |
| Thai | th | M | 84.5 | 44.7 | 11.5 |
| Bulgarian | bg | M | 88.7 | 59.7 | 44.6 |
| Hindi | hi | M | 85.3 | 48.8 | 5.6 |
| Urdu | ur | L | 82.9 | 43.7 | 6.3 |
| Swahili | sw | X | 83.4 | 50.3 | 40.8 |
| Average | | | 87.1 | 57.0 | 37.3 |

Table 5: Accuracy of ChatGPT (zero-shot learning) and mT5-XXL (supervised learning with English and translated data) on the development set of XNLI. ChatGPT is evaluated with both English (en) and language-specific (spc) task descriptions.

and mT5-XXL also seem smaller for high-resource languages. Finally, ChatGPT with target-language task descriptions produces significantly lower accuracy than those with English task descriptions across all considered languages, suggesting the benefits of English descriptions for multilingual NLI with ChatGPT.

## 8 Question Answering

Given a context passage and a question, a Question Answering (QA) model needs to return the answer for the question, which should be a span of text in the input passage. To this end, we utilize the XQuAD dataset (Artetxe et al., 2020) to evaluate ChatGPT in multiple languages for QA. XQuAD involves 240 paragraphs and 1190 question-answer pairs in English and their translations into ten other languages for evaluation. We describe our Chat-GPT prompt for QA in Appendix F.

Given the responses from ChatGPT for our QA prompts for the examples, we remove the period characters in the end and directly evaluate remaining responses using the SQuAD's scorer[5], which is suggested by the original paper of XQuAD (Artetxe et al., 2020).

[5] https://raw.githubusercontent.com/allenai/bi-att-flow/master/squad/evaluate-v1.1.py

| Language | Code | Cat. | mT5-XXL | | ChatGPT(en) | | ChatGPT(spc) | |
|---|---|---|---|---|---|---|---|---|
| | | | EM | F1 | EM | F1 | EM | F1 |
| English | en | H | 80.3 | 91.3 | 56.0 | 74.9 | 56.0 | 74.9 |
| Russian | ru | H | 70.4 | 85.2 | 30.2 | 49.1 | 22.4 | 52.6 |
| German | de | H | 68.2 | 85.0 | 45.9 | 65.8 | 44.7 | 65.8 |
| Chinese | zh | H | 80.0 | 85.7 | 37.1 | 42.3 | 20.5 | 20.8 |
| Spanish | es | H | 70.8 | 87.4 | 41.8 | 65.8 | 40.5 | 69.1 |
| Vietnamese | vi | H | 67.1 | 85.3 | 36.1 | 57.3 | 26.8 | 60.8 |
| Turkish | tr | M | 67.7 | 84.4 | 34.5 | 56.4 | 18.3 | 52.8 |
| Arabic | ar | M | 68.2 | 83.4 | 32.0 | 50.3 | 24.1 | 49.9 |
| Greek | el | M | 68.9 | 85.9 | 29.7 | 45.0 | 17.7 | 39.1 |
| Thai | th | M | 74.5 | 80.2 | 31.2 | 43.4 | 1.5 | 13.1 |
| Hindi | hi | M | 68.2 | 83.7 | 17.5 | 37.8 | 0.6 | 22.9 |
| Average | | | 71.3 | 85.2 | 35.6 | 53.5 | 21.7 | 47.4 |

Table 6: Performance of ChatGPT (zero-shot learning) and mT5-XXL (supervised learning with translated data) on the XQuAD dataset. en and spc indicate whether ChatGPT uses English or target language prompts. The performance is computed using exact match (EM) and F1 scores.

**Results:** Table 6 shows the performance of Chat-GPT (zero-shot learning) and mT5-XXL (Xue et al., 2021), a state-of-the-art supervised learning model for XQuAD. As such, for each language, mT5-XXL is trained over the combination of English training data and the translations to the target language to achieve optimal performance. We report the performance using both the exact match (EM) and F1 scores. Table 6 illustrates that Chat-GPT's zero-shot performance is significantly worse than the supervised model mT5-XXL for all the languages. Across different models and prompts, the QA performance for English is significantly better than those for other languages, demonstrating the clear bias for English of current multilingual language models. Finally, we find that prompting ChatGPT with English tends to produce better performance for multilingual QA than using target languages.

## 9 Common Sense Reasoning

Common Sense Reasoning (CSR) evaluates the reasoning of the models via multiple-choice questions. The inputs for the models involve a question and a few choices for the answer, and the models need to select one of the choices. To evaluate ChatGPT's multilingual abilities for CSR, we leverage two datasets: (i) X-CSQA (Talmor et al., 2019; Lin et al., 2021), which involves English data and its translations to 15 other languages, and (ii) Wikipedia Cloze QA from IndicNLPSuite (Kakwani et al., 2020), which covers 11 low- and extremely-low-resource Indian languages. We evaluate the models on the dev set of X-CSQA with 1,000 samples for each language, while the Wiki

| Language | Code | Cat. | TRT | ChatGPT | |
| --- | --- | --- | --- | --- | --- |
| | | | | (en) | (tgt) |
| English | en | H | 70.0 | 75.0 | 75.0 |
| Russian | ru | H | 59.8 | 50.2 | 53.5 |
| German | de | H | 61.7 | 52.6 | 61.0 |
| Chinese | zh | H | 59.6 | 50.2 | 42.5 |
| Japanese | jp | H | 54.3 | 41.9 | 43.0 |
| French | fr | H | 60.9 | 50.5 | 61.7 |
| Spanish | es | H | 61.1 | 53.3 | 62.5 |
| Italy | it | H | 61.2 | 50.6 | 55.9 |
| Dutch | nl | H | 59.8 | 52.9 | 60.4 |
| Polish | pl | H | 59.7 | 35.2 | 51.1 |
| Portugese | pt | H | 60.5 | 49.5 | 59.2 |
| Vietnamese | vi | H | 59.3 | 42.3 | 47.9 |
| Arabic | ar | M | 58.1 | 49.4 | 47.3 |
| Hindi | hi | M | 53.8 | 41.1 | 38.6 |
| Urdu | ur | L | 52.8 | 34.7 | 24.5 |
| Swahili | sw | X | 51.8 | 35.6 | 46.6 |
| Average | | | 59.0 | 47.8 | 51.9 |

Table 7: Accuracy of ChatGPT (zero-shot learning) and TRT (supervised learning) on the dev set of X-CSQA dataset. en and spc indicate whether ChatGPT uses English or language-specific prompts.

Cloze QA dataset from IndicNLPSuite contains 62,314 samples for all languages. Appendix G presents our ChatGPT prompt for CSR.

**Results**: Table 7 reports the accuracy of Chat-GPT (zero-shot learning for both English and language-specific prompts) and the state-of-the-art supervised model TRT (Fang et al., 2022) on the X-CSQA dataset. TRT is based on the XLM-RoBERTa large model (Conneau et al., 2020) where commonsense knowledge in different sources is retrieved to enrich input questions and answers. Except for English, the table illustrates the poorer performance of ChatGPT than TRT across all other languages for CSR on X-CSQA when the English task description is used. Interestingly, in contrast to other tasks, we find that language-specific prompts tend to perform better than English prompts for ChatGPT in CSR for high-resource languages (except for Chinese), leading to some improvement over supervised learning (e.g. for French, Spanish, and Dutch).

For IndicNLPSuite, Table 8 demonstrates the accuracy of ChatGPT and IndicBERT (Kakwani et al., 2020), a pre-trained encoder-only model using the ALBERT architecture over an Indian language corpora. IndicBERT is fine-tuned on training data to deliver state-of-the-art performance for IndicNLP-

Suite in the original paper (Kakwani et al., 2020). Our experiment results for IndicNLPSuite confirm the general tendency that supervised learning models still perform better than ChatGPT over different languages. However, there are two exceptions with Hindi and Kannada where ChatGPT can produce better accuracy over IndicNLPSuite. Finally, Table 8 suggests that English prompts are a better way to prompt ChatGPT for Indian languages than these languages themselves (except for Marathi and Gujarati).

| Language | Code | Cat. | Indic-BERT | ChatGPT | |
| --- | --- | --- | --- | --- | --- |
| | | | | (en) | (tgt) |
| Hindi | hi | M | 41.6 | 45.7 | 45.2 |
| Bengali | bn | L | 39.4 | 35.2 | 22.0 |
| Tamil | ta | L | 31.8 | 27.9 | 22.3 |
| Malayalam | ml | L | 35.4 | 32.2 | 14.3 |
| Marathi | mr | L | 44.9 | 36.2 | 36.9 |
| Telugu | te | L | 32.6 | 32.5 | 22.2 |
| Gujarati | gu | L | 70.8 | 15.2 | 25.8 |
| Kannada | kn | L | 39.6 | 42.0 | 12.9 |
| Punjabi | pa | X | 44.7 | 38.1 | 27.9 |
| Odia | or | X | 39.3 | 34.7 | 32.9 |
| Assamese | as | X | 40.5 | 35.2 | 24.8 |
| Average | | | 41.1 | 34.1 | 26.1 |

Table 8: Accuracy of ChatGPT (zero-shot learning) and IndicBERT (supervised learning) on the Wikipedia Cloze QA dataset (IndicNLPSuite). en and spc indicate whether ChatGPT uses English or language-specific prompts.

Finally, our ChatGPT evaluation for multilingual summarization is included in Appendix H.

## 10 Discussion

The most important findings from our experiment results is that ChatGPT exhibits significantly worse performance than state-of-the-art supervised models for most of considered NLP tasks in different languages. Given the huge costs to train ChatGPT and similar LLMs as well as the necessity of paid APIs to run large amounts of requests with OpenAI, it seems more reasonable to build smaller task-specific models for NLP problems (or at least for the considered tasks) in different languages that can be hosted locally to serve at lower costs.

In addition, we notice an exception for the POS tagging task where ChatGPT can achieve competitive or even better performance than the supervised learning models (especially with English prompts) over different languages. For instance,

ChatGPT has significantly better POS tagging accuracy for Thai, Vietnamese, Bulgarian, Hindi, and Urdu, which are medium- and low-resource languages. As such, in contrast to other considered tasks which require some level of semantic reasoning, POS tagging focuses on low-level syntactic analysis. We thus hypothesize that ChatGPT possesses high-level skills in grammar and low-level abilities of semantic reasoning to generate seemingly fluent texts for multiple languages. However, for more complicated semantic analysis, ChatGPT might find it more challenging to perform accurate predictions and generations.

Regarding the classification of high-, medium-, low-, and extremely low-resource languages, our work currently relies on data ratios for the languages in the CommonCrawl corpus. According to our experiments, it is interesting that the performance of ChatGPT for low- and extremely-low-resource languages in some tasks is better or comparable to those for high- or medium-resource languages. For instance, for POS tagging in Table 2, ChatGPT's performance for Urdu (a low-resource language) is better than the performance for Vietnamese and Thai (high- and medium-resource languages). In NER, ChatGPT achieves better performance for the low-resource language Bengali than for Chinese (using English prompts in Table 3). For the common sense reasoning task in Table 7, ChatGPT's performance for the extremely-low-resource language Swahili is comparable to those for Polish (with English prompts). To this end, it seems evident that data size might not be the only factor that dictates the resource level and performance for a task of a language with ChatGPT and LLMs.

Compared to language-specific prompts, the superior performance of ChatGPT with English task descriptions over a majority of problems and languages suggests that ChatGPT might better understand/analyze the tasks with English prompts to lead to improved abilities to generate responses with accurate outputs. In addition, the inclusion of English task descriptions for non-English inputs can be seen as an approach to shift the representations of language-specific inputs toward the English space that can be better processed by ChatGPT due to the domination of English in its training data. However, we also note some recent work that reveals a rather different findings, suggesting that ChatGPT can perform competitively or even better

with language-specific prompts for NLP tasks in target languages (Hasan et al., 2023; Deng et al., 2023). A reason for those different findings might come from potentially different versions of ChatGPT at different times that are used to conduct the studies. It thus highlights the importance of better transparency for LLMs, e.g., with respect to training data (Nguyen et al., 2023), to allow accurate and deeper investigation of the models. Finally, the better performance with English prompts also raises an interesting question on whether English is the optimal language to prompt ChatGPT or it is better to employ other languages for this purpose for different target languages.

## 11   Conclusion

Toward a more comprehensive understanding of ChatGPT and LLMs on their multilingual learning abilities for NLP, our work conducts an evaluation for ChatGPT on 7 different tasks, i.e., Part-of-Speech Tagging, Named Entity Recognition, Relation Extraction, Natural Language Inference, Question Answering, Common Sense Reasoning, and Summarization. Using 37 diverse languages with high-, medium-, low-, and extremely low resources for the experiments, our results reveal the less optimal performance of ChatGPT in the zero-shot learning setting for NLP tasks in different languages, advocating for task-specific models to secure best performance. As an ongoing research, we plan to extend the experiments to include more languages, tasks, models, criteria, and settings in future work to obtain broader and deeper insights.

## Acknowledgement

This research has been supported by the Army Research Office (ARO) grant W911NF-21-1-0112, the NSF grant CNS-1747798 to the IUCRC Center for Big Learning, and the NSF grant # 2239570. This research is also supported in part by the Office of the Director of National Intelligence (ODNI), Intelligence Advanced Research Projects Activity (IARPA), via the HIATUS Program contract 2022-22072200003. The views and conclusions contained herein are those of the authors and should not be interpreted as necessarily representing the official policies, either expressed or implied, of ODNI, IARPA, or the U.S. Government. The U.S. Government is authorized to reproduce and distribute reprints for governmental purposes notwithstanding any copyright annotation therein.

## Limitations

As an ongoing work to evaluate ChatGPT and LLMs on multilingual learning tasks, our current work observes several limitations that can be addressed in future studies. First, although our experiments have covered 37 languages, including low- and extremely low-languages, there are still many other languages that are not explored in the current work. Some tasks/datasets in our work have not covered lower-resource languages. The future work can expand the language set with greater focuses on lower-resource languages to better understand LLMs' performance in this important direction. Second, many other tasks, including those with available multilingual datasets, have not been considered in the current work. Examining more tasks and datasets will enable a more comprehensive understanding of ChatGPT and LLMs in multilingual settings. Third, our current work only evaluates ChatGPT in the zero-shot learning setting, thus unable to show comparisons with other recent multilingual LLMs, e.g., BLOOM (Scao et al., 2022), GPT-4, and BARD, in various learning scenarios. While some of these models are currently less accessible for large-scale evaluations, our plan is to further include more models and learning settings along the way to strengthen our evaluations and comparisons when possible. Finally, the current work only evaluates ChatGPT in terms of performance over NLP tasks in different languages. To better characterize ChatGPT and LLMs, other evaluation metrics should also be investigated to report more complete perspectives for multilingual learning, including but not limited to adversarial robustness, biases, toxic/harmful content, hallucination, accessibility, development costs, and interpretability.

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

**Task Description:** Please provide the POS tags for each word in the input sentence. The input will be a list of words in the sentence. The output format should be a list of tuples, where each tuple consists of a word from the input text and its corresponding POS tag label from the tag label set: ["ADJ", "ADP", "ADV", "AUX", "CCONJ", "DET", "INTJ", "NOUN", "NUM", "PART", "PRON", "PROPN", "PUNCT", "SCONJ", "SYM", "VERB", "X"].
**Note:** Your response should include only a list of tuples, in the order that the words appear in the input sentence, with each tuple containing the corresponding POS tag label for a word.
**Input:** ["What", "if", "Google", "Morphed", "Into", "GoogleOS", "?"]

⇒ *[("What", "PRON"), ("if", "SCONJ"), ("Google", "PROPN"), ("Morphed", "VERB"), ("Into", "ADP"), ("GoogleOS", "PROPN"), ("?", "PUNCT")].*

Figure 1: Input prompt and output of ChatGPT for the XGLUE-POS dataset.

## A  Part-of-Speech Tagging Prompt

Our prompt for POS tagging for ChatGPT consists of a task description, a note for output format, and an input sentence, concatenated in that order, i.e., $Prompt_{POS}$ = [*task description; output format note; input sentence*]. Notably, instead of directly using the text of input sentence, we feed ChatGPT with the list of words in the sentence to facilitate the word-label alignment and parsing of ChatGPT responses for POS tagging. Our task description and output format note then emphasize on the expected format for the ChatGPT's responses to follow the tuple structure with pairs of words and their corresponding POS tags. In the experiments, this approach has led to better performance for ChatGPT than the direct input sentence. We illustrate an example for the English POS prompts for ChatGPT in Figure 1.

## B  Named Entity Recognition Prompt

Our prompt structure for ChatGPT with Named Entity Recogntion (NER) follows the prompts for POS Tagging, i.e., $Prompt_{NER}$ = [*task description; output format note; input sentence*], which involve a task description to explain the task and list entity type/labels of interest. We also have a note to

specify the expected output format with tuples of words and predicted tags for names. However, a key difference for NER is that we explicitly ask ChatGPT to produce tags for each work in the BIO format. Although this approach seems to make the task more challenging for ChatGPT, we find that it actually improves the performance for ChatGPT. Our hypothesis is that the BIO tag requirement encourages ChatGPT to solve NER as a sequence labeling problem, thus forcing it to comprehensively annotate names in input sentences. In contrast, the simpler approach to prompt ChatGPT for names without BIO specification might suggest reading comprehension formulation that does not tag all names with exact spans for NER. The responses from ChatGPT are also harder (i.e., more ambiguous and unpredictable) to parse for NER outputs without the BIO requirement. We provide an English prompt example for NER for ChatGPT in Figure 2.

**Task Description:** You are working as a named entity recognition expert and your task is to label a given text with named entity labels. Your task is to identify and label any named entities present in the text. The named entity labels that you will be using are PER (person), LOC (location), CORP (corporation), CW (creative work), GRP (group of people), and PROD (product). You may encounter multi-word entities, so make sure to label each word of the entity with the appropriate prefix ("B" for the first word of the entity, "I" for any non-initial word of the entity). For words which are not part of any named entity, you should return "O".
**Note:** Your output format should be a list of tuples, where each tuple consists of a word from the input text and its corresponding named entity label.
**Input:** ["john", "is", "first", "mentioned", "in", "a", "charter", "from", "1247", "."]

⇒ *[("john", "B-PER"), ("is", "O"), ("first", "O"), ("mentioned", "O"), ("in", "O"), ("a", "O"), ("charter", "B-CW"), ("from", "O"), ("1247", "B-PROD"), (".", "O")].*

Figure 2: Input prompt and output of ChatGPT for the MultiCoNER dataset.

## C Analysis of ChatGPT's Performance for Named Entity Recognition

| Label | Precision | Recall | F1 | Spurious (%) |
|-------|-----------|--------|------|-----|
| CORP | 31.0 | 33.8 | 32.1 | 39 |
| CW | 12.0 | 17.2 | 14.1 | 57 |
| GRP | 6.7 | 5.7 | 6.2 | 26 |
| LOC | 33.2 | 37.7 | 34.9 | 44 |
| PER | 51.2 | 66.1 | 57.5 | 32 |
| PROD | 20.3 | 22.3 | 21.1 | 55 |

Table 9: ChatGPT label-wise scores on MultiCoNer

In order to better understand the performance of ChatGPT for MultiCoNER, we use the scoring script *nervaluate*[6] to compute detailed scores for each entity types for ChatGPT. Table 9 shows label-wise precision, recall, and F1 scores of ChatGPT (with English prompts). We also include spurious percentages (over total numbers of predictions), which are the percentages of ChatGPT's predictions that do not exist in the annotated data for each type. As can be seen, ChatGPT's extraction performance is very poor for GRP (group of people) and CW (creative work), which have F1 scores of less than 15%. Also, the spurious percentages of Chat-GPT are generally high for all entity types, which suggests ChatGPT's verbosity and confusion for NER.

## D Relation Extraction Prompt

An input example for RE involves an input text and two entity mentions in the text for classification. To probe ChatGPT for RE for an example, we design the prompt via the concatenation of a task description, input text, and two entity mentions, i.e., $Prompt_{RE}$ = [*task description; output format note; input text; entity 1; entity 2*]. In the task description for RE, we explicitly include all the relation types to inform ChatGPT. We also introduce an output format note to specify the expected format for the responses from ChatGPT for RE, thus facilitating response parsing for relation labels. To illustrate the RE prompts for ChatGPT, we present an example with the English prompt and corresponding response in Figure 3.

[6] https://github.com/MantisAI/nervaluate

**Task Description:** Given a input text describing the relationship between two entities, extracts the relationship between them. The relation has to be of the type: "birth-place", "eats", "event-year", "first-product", "from-country", "has-author", "has-child", "has-edu", "has-genre", "has-height", "has-highest-mountain", "has-length", "has-lifespan", "has-nationality", "has-occupation", "has-parent", "has-population", "has-sibling", "has-spouse", "has-tourist-attraction", "has-type", "has-weight", "headquarters", "invented-by", "invented-when", "is-member-of", "is-where", "loc-leader", "movie-has-director", "no-relation", "org-has-founder", "org-has-member", "org-leader", "post-code", "starring", "won-award".
**Note:** Your output must only be the relation of the two given entities and must follow the format: "Relation: <One of the above listed relations>".
**Input:** North West Coastal Highway is a generally north-south Western Australian highway which links the coastal city of Geraldton with the town of Port Hedland.
**Entity 1:** North West Coastal Highway
**Entity 2:** highway

⇒ *Relation: has-type.*

Figure 3: Input and output of ChatGPT for the SMiLER dataset.

## E Natural Language Inference Prompt

To construct the prompt for ChatGPT for each example in XNLI, we directly concatenate the task description, the premise, the hypothesis, and a multiple choice question (of entailment, contradiction, and neural) in this order, i.e., $Prompt_{NLI}$ = [*task description; premise; hypothesis; question*]. An example of English input prompts and responses from ChatGPT is shown in Figure 4.

## F Question Answering Prompt

We collect the English task description for QA from the NaturalInstructions repository (Wang et al., 2022b) for ChatGPT. In addition, as ChatGPT tends to generate long responses, we introduce a note to remind the model that the answers for our dataset should be short and directly extracted from the input passage. This approach has helped ChatGPT

**Task Description:** Please identify whether the premise entails or contradicts the hypothesis in the following premise and hypothesis. The answer should be exact "entailment", "contradiction", or "neutral".
**Premise:** And he said, Mama, I'm home.
**Hypothesis:** He called his mom as soon as the school bus dropped him off.
Is it entailment, contradiction, or neutral?

⇒ *Neutral. The premise doesn't confirm or deny the hypothesis....*

Figure 4: Input prompt and output of ChatGPT for the XNLI dataset.

to provide more direct answers in our experiments. To this end, for an example with an input passage and question, our prompt for ChatGPT is formed via: $Prompt_{QA}$ = [*task description; passage; question; note*]. We demonstrate an example of the QA prompts in Figure 5.

## G  Common Sense Reasoning Prompt

In the CSR prompts for ChatGPT, we combine the task description, the question, and the multiple choices for each sample, i.e., $Prompt_{CSR}$ = [*task description; question; multiple choices*]. Here, for the task description, we also indicate the language of the input question and multiple choices. Two examples of prompts for CSR inputs are presented in Figure 6 for the X-CSQA dataset and in Figure 7 for the Wikipedia Cloze QA dataset from IndicNLPSuite.

## H  Summarization

In summarization, systems need to provide key and concise information for a longer input text, which can be helpful for different downstream applications such as news analysis, marketing, question answering, and scientific document processing. To study the performance of ChatGPT for summarization in multiple languages, we choose the XL-Sum dataset (Hasan et al., 2021) that provides summaries of news articles in 44 languages. In contrast to extractive summarization that select important sentences in the input text to a summary, XL-Sum addresses abstractive summarization to allow text generation with more creative writing in the summary (the sentences in the summary might not necessarily appear in the input text). Despite

**Task Description:** Answer the question from the given passage. Your answer should be directly extracted from the passage, and it should be a single entity, name, or number, not a sentence.
**Passage:** Peyton Manning became the first quarterback ever to lead two different teams to multiple Super Bowls. He is also the oldest quarterback ever to play in a Super Bowl at age 39. The past record was held by John Elway, who led the Broncos to victory in Super Bowl XXXIII at age 38 and is currently Denver's Executive Vice President of Football Operations and General Manager.
**Question:** How old was Peyton Manning when he played in Super Bowl 50?
**Note:** Your answer should be directly extracted from the passage and be a single entity, name, or number, not a sentence.

⇒ *39.*

Figure 5: Input prompt and output of ChatGPT for XQUAD dataset.

**Task description:** In this task, you will be presented with a question that has multiple possible answers in English. You should choose the most suitable option out of "A", "B", "C", "D", and "E", based on your commonsense knowledge.
**Question:** When you return to work you will likely need what to get in the door if you are the first to arrive?
**Options**:
A earn money
B key
C need money
D badge
E get out of bed

⇒ *Option B is the most suitable answer: key.*

Figure 6: Input prompt and output of ChatGPT for X-CSQA dataset.

greater challenges, abstractive summarization can produce more natural texts to better serve downstream applications.

To facilitate the experiments, we select 12 languages in XL-Sum, covering high-, medium-, low-, and extremely low-resource languages, and eval-

**Task description:** You are given a statement written in Hindi. Choose the most logical word from the 4 given options which can be used to replace the <MASK> token in the statement. Output the word from the correct option .

**Statement:** रतन देवासी का जन्म 25 सितंबर 1975 को <MASK> के सिरोही जिले में माउंट आबू में हुआ था। इनके पिता का नाम शंकरलाल देवासी तथा पत्नी का नाम विराज देवासी है। देवासी बचपन से ही मेधावी छात्र रहे है। वे डिप्लोमा इन होटल मैनेजमेंट डिग्रीधारक है। देवासी छात्र जीवन से ही तेज-तर्रार एवं मृदुभाषी है।
Option A: कांग्रेस
Option B: एनएसयूआई
Option C: राजस्थान
Option D: लोकसभा

⇒ *Option C:* राजस्थान

Figure 7: Input prompt and output of ChatGPT for Wikipedia Cloze QA dataset (IndicNLPSuite). Translation of the statement and options by Google Translate: *Ratan Devasi was born on 25 September 1975 at Mount Abu in the Sirohi district of <MASK>. His father's name is Shankarlal Devasi and wife's name is Viraj Devasi. Devasi has been a brilliant student since childhood. He is a Diploma in Hotel Management degree holder. Devasi is quick-tempered and soft-spoken since his student life. Option A: Congress; Option B: NSUI; Option C: Rajasthan; Option D: Lok Sabha.*

uate ChatGPT's performance on the test datasets of the languages. Table 10 shows the sizes of test data (i.e., the numbers of samples) in XL-Sum for the selected languages. In the experiments, we utilize the ROUGE-1, ROUGE-2, and ROUGE-L scores as performance measures for summarization. Note that for the non-English languages, the scorer script in the original paper of XL-Sum (Hasan et al., 2021) is used for performance computation.

As a summary in XL-Sum is expected to be written in the same language as the input text, given an input text, our summarization prompt for ChatGPT is constructed via the concatenation: $Prompt_{SUM}$ = [*task description; output language specification: input text*]. Accordingly, the task description is simply: "*Summarize this* <lang> *text.*" while the output langauge specification is expressed via: "*The output should be in* <lang>". Here, <lang> indicates the the same language that is presented in the input text and expected in the summary

response. <lang> can be translated into appropriate languages as required by the language of the prompts. For instance, using English for the prompts, the summarization prompt for a French input is "*Summarize this French text. The output should be in French: . . .*". In the experiments, we find that ChatGPT might generate responses in English even for non-English inputs and including output language specifications in the prompts is important to instruct the same language in the inputs and outputs for ChatGPT.

**Results:** Tables 10 and 11 presents the summarization performance of ChatGPT (zero-shot learning) for the selected languages in XL-Sum using English and language-specific prompts respectively. In the tables, we also include the performance of the mT5-XXL model that is trained over training data of specific languages in XL-Sum. mT5-XXL has achieved state-of-the-art performance for XL-Sum as reported in (Aharoni et al., 2022). It is obvious from the tables that ChatGPT's performance is consistently inferior to mT5-XXL's with large performance gaps in different languages. To better understand the poor performance of ChatGPT, Tables 10 and 11 also report the average lengths of the human-provided summaries and the summaries generated by ChatGPT (in terms of the numbers of characters). It is clear from the tables that ChatGPT tends to generate lengthy summaries, potentially leading to its poorer performance. In addition, the tables show the success rates of ChatGPT for each language, which is defined as the ratios of requests sent to the ChatGPT server and received non-empty responses/summaries. As can be seen, the success rates of ChatGPT for lower-resource languages are also lower that can further explain ChatGPT's performance and reliability for such languages.

| Language | Code | Cat. | Size | ChatGPT | | | Avg. Gold Length | Avg. Model Length | Success (%) | mT5-XXL L |
|---|---|---|---|---|---|---|---|---|---|---|
| | | | | **1** | **2** | **L** | | | | |
| English | en | H | 11,535 | 19.71 | 5.52 | 13.38 | 125.84 | 612.38 | 99 | 32.51 |
| Russian | ru | H | 7,780 | 18.65 | 5.13 | 12.83 | 182.11 | 523.03 | 96 | 28.48 |
| Chinese | zh | H | 4,670 | 21.14 | 5.31 | 15.27 | 420.10 | 191.46 | 98 | 33.54 |
| French | fr | H | 1,086 | 20.76 | 7.09 | 14.12 | 147.38 | 601.17 | 99 | 34.12 |
| Spanish | es | H | 4,763 | 17.81 | 4.44 | 11.97 | 163.39 | 719.52 | 99 | 27.40 |
| Turkish | tr | M | 3,397 | 14.52 | 4.54 | 10.87 | 164.83 | 610.75 | 99 | 30.80 |
| Arabic | ar | M | 4,689 | 19.37 | 5.36 | 13.64 | 142.95 | 396.86 | 95 | 32.00 |
| Thai | th | M | 826 | 17.55 | 5.35 | 11.51 | 218.13 | 275.89 | 59 | 30.59 |
| Hindi | hi | M | 8,847 | 21.06 | 5.63 | 14.21 | 137.24 | 294.93 | 82 | 36.88 |
| Bengali | bn | L | 1,012 | 6.34 | 1.63 | 4.65 | 148.19 | 176.08 | 42 | 34.19 |
| Burmese | my | L | 570 | 14.49 | 6.32 | 8.85 | 201.99 | 118.78 | 43 | 41.40 |
| Kyrgyz | ky | X | 500 | 5.10 | 1.47 | 4.16 | 188.14 | 437.51 | 87 | 26.48 |

Table 10: Performance of ChatGPT (zero-shot learning) (ROUGE-1/2/L) and mT5-XXL (supervised learning) (ROUGE-L) for summarization over XL-Sum using English prompts.

| Language | Code | Cat. | Size | ChatGPT | | | Avg. Gold Length | Avg. Model Length | Success (%) | mT5-XXL L |
|---|---|---|---|---|---|---|---|---|---|---|
| | | | | **1** | **2** | **L** | | | | |
| English | en | H | 11,535 | 21.38 | 5.97 | 14.48 | 125.84 | 524.15 | 99 | 32.51 |
| Russian | ru | H | 7,780 | 15.60 | 4.17 | 10.83 | 182.11 | 483.45 | 96 | 28.48 |
| Chinese | zh | H | 4,670 | 11.65 | 2.61 | 8.96 | 55.31 | 420.10 | 98 | 33.54 |
| French | fr | H | 1,086 | 21.11 | 7.21 | 14.49 | 147.38 | 512.17 | 100 | 34.12 |
| Spanish | es | H | 4,763 | 19.73 | 4.85 | 13.15 | 163.39 | 601.05 | 99 | 27.40 |
| Turkish | tr | M | 3,397 | 15.58 | 4.91 | 11.79 | 164.83 | 468.64 | 99 | 30.80 |
| Arabic | ar | M | 4,689 | 16.95 | 4.74 | 12.04 | 142.95 | 383.71 | 94 | 32.00 |
| Thai | th | M | 826 | 14.39 | 4.11 | 9.71 | 218.13 | 257.40 | 58 | 30.59 |
| Hindi | hi | M | 8,847 | 4.28 | 1.13 | 2.94 | 137.24 | 423.58 | 82 | 36.88 |
| Bengali | bn | L | 1,012 | 1.88 | 0.43 | 1.39 | 148.19 | 198.60 | 40 | 34.19 |
| Burmese | my | L | 570 | 0.45 | 0.34 | 0.44 | 201.99 | 152.27 | 40 | 41.40 |
| Kyrgyz | ky | X | 500 | 8.40 | 2.23 | 6.42 | 188.14 | 458.71 | 86 | 26.48 |

Table 11: Performance of ChatGPT (zero-shot learning) (ROUGE-1/2/L) and mT5-XXL (supervised learning) (ROUGE-L) for summarization over XL-Sum using language-specific prompts.