# OpenReview forum: "ChatGPT Beyond English: Towards a Comprehensive Evaluation of Large Language Models in Multilingual Learning"
_EMNLP/2023/Conference — EMNLP 2023 Findings_

### Official Review · Reviewer_6NDZ · 2023-08-03

**Typos Grammar Style And Presentation Improvements:** 1. line 269
**Soundness:** 3

**Excitement:**

3: Ambivalent: It has merits (e.g., it reports state-of-the-art results, the idea is nice), but there are key weaknesses (e.g., it describes incremental work), and it can significantly benefit from another round of revision. However, I won't object to accepting it if my co-reviewers champion it.

**Paper Topic And Main Contributions:**

This paper investigates the effectiveness of large language models (LLMs), specifically ChatGPT, in non-English language tasks. Although ChatGPT has made significant strides in English language processing, there has been limited and insufficient exploration of its performance across other languages. The paper addresses the question of whether ChatGPT can effectively operate in other languages or if there's a need for more language-specific technologies. The authors conduct extensive testing of ChatGPT across seven different tasks and 37 languages, which vary in terms of the availability of linguistic resources.

Their findings reveal several important aspects.

- Firstly, the performance of ChatGPT in zero-shot learning scenarios is generally worse than that of supervised learning models for most tasks across different languages, including those with high, medium, low, and extremely low resources. This underscores the need for more task-specific models in natural language processing.

- Secondly, ChatGPT's performance is better for English than for other languages, particularly in tasks that require complex reasoning abilities, such as named entity recognition, question answering, common sense reasoning, and summarization. This finding suggests that the model has a bias towards English, indicating the necessity for more language-specific models for different languages and groups.

- Lastly, they also found that ChatGPT performs better with English prompts, even when the task and input texts are intended for other languages. This discovery further confirms ChatGPT's English-centric bias.


**Reasons To Accept:**



1. The paper addresses a significant gap in the evaluation of large language models (LLMs) like ChatGPT for diverse languages.

2. It carries out a comprehensive evaluation on 37 diverse languages and 7 different tasks, which is one of the largest studies of its kind.



**Reasons To Reject:**

1. The conclusions of this paper is somewhat not surprising to me, as there is no wonder that most of the languages underperforms English tasks, as English corpora takes the majority of most of LLMs' training.
2. The presentation of this paper does not look like a paper but a technical report. I am not judging the quality of this paper at this point but the content is like putting various experiments into a paper without a clear roadmap. Thus, it makes feel that the authors have too much to convey, but an obvious drawback of this is nothing in depth at the end.


**Reproducibility:**

3: Could reproduce the results with some difficulty. The settings of parameters are underspecified or subjectively determined; the training/evaluation data are not widely available.

**Reviewer Confidence:**

3: Pretty sure, but there's a chance I missed something. Although I have a good feel for this area in general, I did not carefully check the paper's details, e.g., the math, experimental design, or novelty.

---

> ### Author Rebuttal · Authors · 2023-08-28
>
> Thank you for your comments and suggestions. Please find below our responses for your questions.
>
> **Reviewer**: *"The conclusions of this paper is somewhat not surprising to me, as there is no wonder that most of the languages underperforms English tasks ..."*
>
> **Our Response**:
>
> Thank you for your comment. In addition to the general trend of poorer performance of ChatGPT for non-English languages, our study also provides quantitative evaluation to show how significant the performance gaps between English and non-English languages can be. Given ChatGPT's notably better performance in English, our experiments highlight the importance of creating language-specific models to inform future development of large language models. Furthermore, our evaluation illustrates that supervised learning models can still significantly outperform ChatGPT across various downstream tasks. This reaffirms the significance of task-specific models in ensuring optimal performance for different NLP problems.
>
> **Reviewer**: *"The presentation of this paper does not look like a paper but a technical report. ... but the content is like putting various experiments into a paper without a clear roadmap.  ..."*
>
> **Our Response**:
>
> Thank you for your comment. The main goal of our work is to provide a comprehensive understanding of ChatGPT's performance in the multilingual learning settings. This is why a thorough evaluation of ChatGPT for many NLP tasks and languages as done in our work is necessary to obtain sufficient data points to strengthen our exploration. The consistent findings from our experiments regarding ChatGPT's performance offer insightful information to contextualize ChatGPT in the development of NLP models for future research.

---

### Official Review · Reviewer_3x9S · 2023-08-06

**Typos Grammar Style And Presentation Improvements:** This is a well-written and structured…
**Soundness:** 5

**Excitement:**

4: Strong: This paper deepens the understanding of some phenomenon or lowers the barriers to an existing research direction.

**Paper Topic And Main Contributions:**

This work present a thorough evaluation about whether the ability of ChatGPT on NLP tasks can generalise to different languages. Extensively tested with languages with different volumes of available resources, the authors provide several insights for future research:

- ChatGPT is not a general problem solver for NLP tasks compared to supervised models regardless of languages
- Confirm that ChatGPT performs generally better on English tasks (mainly because of the pre-training corpus). The deterioration becomes obvious for tasks requiring reasoning.
- English prompts work better across tasks in different languages.



**Questions For The Authors:**

Question A: will this evaluation be extended to other LLMs? It would be useful to benchmark the multilingual NLP ability among different models varying in model size and training corpus.



**Reasons To Accept:**

- This works provides a comprehensive evaluation of the multilingual NLP capacity for ChatGPT.
  - Comprehensively evaluates an important and timely topic - understanding the abilities, biases and limitations of large language models like ChatGPT for diverse languages.
  - Covers a wide range of languages and NLP tasks to analyse different aspects of ChatGPT from the perspective of multilingual learning.
- Provides an standardised evaluation protocol and benchmarking methodology for better reproducibility.

**Reasons To Reject:**

- The version of ChatGPT is not specified. Since the online ChatGPT model is constantly updated, evaluation and analysis without specific date/version information might cause misleadings in the future research.
- Although the authors claim that using only zero-shot is better for reproducing and fits the general user case, missing the results in few-shot setting could be a slip, given that in-context learning could bring significant improvements.

**Reproducibility:**

4: Could mostly reproduce the results, but there may be some variation because of sample variance or minor variations in their interpretation of the protocol or method.

**Reviewer Confidence:**

4: Quite sure. I tried to check the important points carefully. It's unlikely, though conceivable, that I missed something that should affect my ratings.

---

> ### Author Rebuttal · Authors · 2023-08-28
>
> Thank you for your comments and suggestions. Please find below our responses for your questions.
>
> **Reviewer**: *"The version of ChatGPT is not specified.  ..."*
>
> **Our Response**:
>
> Thank you for your comment. In lines 319-324 of the paper, we specify that our results were obtained from ChatGPT between March 1 and April 5. We hope this information can be helpful for future research on the evolution of ChatGPT.
>
> **Reviewer**: *"... missing the results in few-shot setting could be a slip, given that in-context learning could bring significant improvements."*
>
> **Our Response**:
>
> Thank you for your comment. We agree with the reviewer that few-shot learning evaluation can strengthen our results and findings. Our current work focuses on the zero-shot learning setting to present an extensive multilingual evaluation with multiple tasks and languages for ChatGPT. We will extend our work to include other learning settings in future work.
>
> **Reviewer**: *"will this evaluation be extended to other LLMs? It would be useful to benchmark the multilingual NLP ability among different models varying in model size and training corpus."*
>
> **Our Response**:
>
> Thank you for your suggestion. We do plan to evaluate different LLMs with varying sizes in the multilingual learning scenarios, aiming to provide more comprehensive understanding for this important area of NLP. The LLM community is growing very quickly, with many new models like LLaMa, Falcon, and MPT having been introduced recently. However, the present models are predominantly assessed for English and some popular languages. We hope our work can serve as a growing hub for benchmarking the effectiveness of these models and gauging their advancements for multilingual learning. We will also emphasize on under-studied languages in NLP to encourage the impacts of the technologies to a broader population.

---

### Official Review · Reviewer_PmRm · 2023-08-11

**Typos Grammar Style And Presentation Improvements:** 1. The paper has grammatical errors t…
**Soundness:** 3

**Excitement:**

3: Ambivalent: It has merits (e.g., it reports state-of-the-art results, the idea is nice), but there are key weaknesses (e.g., it describes incremental work), and it can significantly benefit from another round of revision. However, I won't object to accepting it if my co-reviewers champion it.

**Missing References:**

1. Touvron, H., Lavril, T., Izacard, G., Martinet, X., Lachaux, M. A., Lacroix, T., ... & Lample, G. (2023). Llama: Open and efficient foundation language models. arXiv preprint arXiv:2302.13971.
2. Chung, H. W., Hou, L., Longpre, S., Zoph, B., Tay, Y., Fedus, W., ... & Wei, J. (2022). Scaling instruction-finetuned language models. arXiv preprint arXiv:2210.11416.

**Paper Topic And Main Contributions:**

This paper presents the evaluation of 7 different NLP tasks covering 37 languages on ChatGPT model.

**Questions For The Authors:**

1. It would be more accurate to report most of the evaluation metric in F1-score (where applicable) rather than accuracy.
2. BLOOMZ is publicly available for large-scale evaluation. Would be more appropriate to see a comparison between ChatGPT and BLOOMZ.

**Reasons To Accept:**

A large-scale study has been done on ChatGPT performance evaluation for 37 different languages on 7 NLP tasks.

**Reasons To Reject:**

1. L184-195, the authors discussed that ChatGPT’s zero-shot learning is worse than state-of-the-art supervised performances. However, some recent study shows that ChatGPT’s zero-shot learning performances can be significantly changed by a simple change in prompt. There is no mention of prompt engineering done by the authors to claim such a statement.
2. L184-195 and L196-207 represent the same meaning. The authors tried to unnecessarily elaborate the same things differently just to put the paper into the long category.
3. L208-211, the authors claimed that English prompts perform better, but a recent study shows, that prompts in other languages can match the level of English language performance if the prompt is similar to English.

**Reproducibility:**

3: Could reproduce the results with some difficulty. The settings of parameters are underspecified or subjectively determined; the training/evaluation data are not widely available.

**Reviewer Confidence:**

4: Quite sure. I tried to check the important points carefully. It's unlikely, though conceivable, that I missed something that should affect my ratings.

---

> ### Author Rebuttal · Authors · 2023-08-28
>
> Thank you for your comments and suggestions. Please find below our responses for your questions.
>
> **Reviewer**: *"...However, some recent study shows that ChatGPT’s zero-shot learning performances can be significantly changed by a simple change in prompt. There is no mention of prompt engineering done by the authors to claim such a statement."*
>
> **Our Response**:
>
> Thank you for your comment. We provide our specific prompts for each task in the evaluation in the Appendix. We agree with the reviewer that ChatGPT's outputs might vary with changes in the input prompts. However, we would like to emphasize that our designed prompts simulate intuitive and typical approaches that a general user would use to prompt ChatGPT for different NLP tasks. While our research can be strengthened by experimenting with other prompt engineering techniques, our work with large sets of tasks and languages introduces a significant scale of evaluation and analysis, providing comprehensive understanding for the performance of ChatGPT in the multilingual settings. Importantly, our prompts involve instructions to ensure that ChatGPT's outputs would follow some specific formats, e.g., the prompts for the POS and NER tasks in Figures 1 and 2 with the word-label tuple format. As such, our prompts can mitigate the variance in the outputs to improve the reliability of the evaluations. We will clarify these details in our final version.
>
> **Reviewer**: *"L184-195 and L196-207 represent the same meaning. The authors tried to unnecessarily elaborate the same things differently just to put the paper into the long category."*
>
> **Our Response**:
>
> Thank you for your comment. However, we would like to emphasize that **L184-195 and L196-207 are presenting two different findings from our paper**. L184-195 discusses the performance comparison between ChatGPT and task-specific supervised learning models, which highlights the benefits of task-specific models for different NLP tasks. In contrast, L196-207 compares ChatGPT's performance for English and the other languages, which shows large performance gaps for the other languages and suggests the necessity of developing language-specific language models. Given our large-scale evaluation with multiple tasks and languages, we believe a long paper is the most suitable format to accommodate our extensive results and analysis. We hope the reviewer can reconsider this point for our paper.
>
> **Reviewer**: *"L208-211, the authors claimed that English prompts perform better, but a recent study shows, that prompts in other languages can match the level of English language performance if the prompt is similar to English."*
>
> **Our Response**:
>
> Thank you for your comment. We hope the reviewer can provide the mentioned recent study and we will certainly include that in our discussion for the final version. While it's unclear to us on your point for "if the prompt is similar to English", we believe our work can complement the contemporary work in the last few months to provide meaningful data points for our multilingual understanding of ChatGPT and large language models.
>
> **Reviewer**: *"It would be more accurate to report most of the evaluation metric in F1-score (where applicable) rather than accuracy."*
>
> **Our Response**:
>
> We employ the same evaluation metrics as done by the state-of-the-art supervised learning models for the tasks to ensure fair comparison. In particular, we provide the F1 scores of the models for the named entity recognition, relation extraction, and question answering tasks. We use the accuracy scores for natural language inference, common sense reasoning (as multiple choice questions), and POS tagging. For summarization, we report the ROGUE scores. We will consider F1 scores for the tasks if applicable in the final version.
>
> **Reviewer**: *"BLOOMZ is publicly available for large-scale evaluation. Would be more appropriate to see a comparison between ChatGPT and BLOOMZ."*
>
> **Our Response**:
>
> Thank you for your comment. As indicated in the Limitations section, extending our experiments to include other large language models would be an important direction for future work. In this study, we focus on measuring the multilingual performance of ChatGPT thoroughly, presenting an extensive evaluation with many tasks and languages. As ChatGPT has garnered considerable public attention with impressive capabilities, our findings can provide insights for the future advancement of large language models (LLMs). These insights emphasize the poor multilingual performance of LLMs if they are trained on data biased towards English and the unfit of LLMs as a general solver for different NLP tasks.
>
> Also, thank you for your suggestions for the additional references. We will discuss them in our final version.

---

### Meta-Review · Area_Chair_U4xf · 2023-09-24

**Recommendation:** 4

**Metareview:**

The authors in this paper study the effectiveness of ChatGPT for multiple tasks in a diverse set of non-English languages.

Pros:
* The reviewers have found the paper is well written and very interesting to read.
* The paper is the first to study and report ChatGPT’s performance on 37 different languages on 7 NLP tasks.
* It is not surprising but the findings that the ChatGPT demonstrates poor performance on various non-English tasks and languages, will be valuable to the community.

Cons:
* The reviewers have pointed out that the paper will significantly improve with inclusion of results from different LLMs and LLMs with varying sizes. While I agree with the reviewers, I believe that the large-scale multilingual analysis of ChatGPT will be valuable to the community.

---

### Decision · Program_Chairs · 2023-10-07

**Decision:**

Accept-Findings

**Comment:**

The authors in this paper study the effectiveness of ChatGPT for multiple tasks in a diverse set of non-English languages.

Pros:
* The reviewers have found the paper is well written and very interesting to read.
* The paper is the first to study and report ChatGPT’s performance on 37 different languages on 7 NLP tasks.
* It is not surprising but the findings that the ChatGPT demonstrates poor performance on various non-English tasks and languages, will be valuable to the community.

Cons:
* The reviewers have pointed out that the paper will significantly improve with inclusion of results from different LLMs and LLMs with varying sizes. While I agree with the reviewers, I believe that the large-scale multilingual analysis of ChatGPT will be valuable to the community.